# Characteristics of Metallic Nanoparticles (Especially Silver Nanoparticles) as Anti-Biofilm Agents

**DOI:** 10.3390/antibiotics13090819

**Published:** 2024-08-28

**Authors:** Hongze Li, Zhihe Yang, Sadaf Aiman Khan, Laurence J. Walsh, Chaminda Jayampath Seneviratne, Zyta M. Ziora

**Affiliations:** 1School of Chemistry and Molecular Biosciences, The University of Queensland, St. Lucia, QLD 4072, Australia; hongze.li@uq.net.au (H.L.); zhihe.yang@uq.net.au (Z.Y.); 2Oral Health Centre, School of Dentistry, The University of Queensland, Herston, QLD 4006, Australia; sadaf.khan@imb.uq.edu.au (S.A.K.); l.walsh@uq.edu.au (L.J.W.); jaya.seneviratne@uq.edu.au (C.J.S.); 3Institute for Molecular Bioscience, The University of Queensland, St. Lucia, QLD 4072, Australia; 4Indian Institute of Technology (IITD) Delhi, New Delhi 110016, India

**Keywords:** antimicrobial agents, biofilm, metallic nanoparticles, bimetallic nanoparticles, silver nanoparticles

## Abstract

Biofilm-associated infections account for a large proportion of chronic diseases and pose a major health challenge. Metal nanoparticles offer a new way to address this problem, by impairing microbial growth and biofilm formation and by causing degradation of existing biofilms. This review of metal nanoparticles with antimicrobial actions included an analysis of 20 years of journal papers and patent applications, highlighting the progress over that time. A network analysis of relevant publications showed a major focus on the eradication of single-species biofilms formed under laboratory conditions, while a bibliometric analysis showed growing interest in combining different types of metal nanoparticles with one another or with antibiotics. The analysis of patent applications showed considerable growth over time, but with relatively few patents progressing to be granted. Overall, this profile shows that intense interest in metal nanoparticles as anti-biofilm agents is progressing beyond the confines of simple laboratory biofilm models and coming closer to clinical application. Looking to the future, metal nanoparticles may provide a sustainable approach to combatting biofilms of drug-resistant bacteria.

## 1. Introduction

A common treatment for bacterial infections is the administration of an antibiotic. However, once the bacteria have organized themselves into a biofilm, the altered metabolic state within the biofilm reduces the effectiveness of antibiotics and makes the emergence or resistance more likely. Additionally, the extracellular polymeric substances (EPSs) of the biofilm limit the penetration of large molecules such as antibiotics and certain biocides, making those ineffective. Given these challenges, there is interest in using metallic nanoparticles with antimicrobial properties to eradicate biofilms or prevent them from forming. The high surface-area-to-volume ratio and ability to penetrate into microbial biofilms make metal nanoparticles attractive as possible therapeutic agents.

According to the US Centers for Disease Control, up to two-thirds of common bacterial infections involve multiple species [1], with more than one type of pathogenic bacteria cooperating with others to form a biofilm [1,2]. In chronic bacterial infections, biofilms may be found in up to 80% of cases [3]. In the biofilm state, the emergence of antimicrobial resistance (AMR) is more likely, making antibiotics and traditional biocides ineffective [4]. 

### Biofilm Formation and the Interactions between Biofilm Components and Antimicrobial Agents

In the biofilm community, bacteria are embedded in EPSs [5], which is composed of polysaccharides, glycopeptides, DNA, lipids, and proteins. Voids in the EPSs serve as channels for the movement of water, nutrients, and waste products [6]. The EPSs protect bacteria from various physical, chemical, biological, and environmental factors, including extremes of temperature and pH, and host immune responses [7]. As a result, biofilms are not easily disrupted.

Biofilms form from planktonic, autotrophic, and heterotrophic bacteria and archaea, typically with more heterotrophic bacteria than autotrophic bacteria [8]. Once individual microbes attach to a surface, they transition from the free-floating state and produce EPSs to anchor them to the surface and to provide protection from the environment [9] (Figure 1). Adhesion and colonization are then followed by maturation [6]. Bacteria transition from reversible adhesion to the surface using van der Waals forces [10] to form stable anchorage, using adhesins and fimbriae. The growth of the biofilm is coordinated through quorum sensing (QS) [11]. From a mature biofilm, planktonic cells can be dispersed into the local environment, to then form biofilms in new locations [7].

Inter-species interactions are essential within the biofilm for its overall persistence and for the survival of its inhabitants, and the term “Sociomicrobiology” coined in 2005 captures this cooperation between inhabitants [12], which may include QS, signaling molecule transmission, and horizontal gene transfer. Cooperation is beneficial for interacting bacteria, providing the foundation for synergy [13]. 

Traditional methods used to treat bacterial and fungal biofilms use physical, chemical, or biological agents. Physical agents include ultraviolet light and ionizing radiation, both of which are dangerous for humans and animals. Biocides as chemical agents can pose a risk to health from damage to healthy host tissues [14], while antibiotics can give rise to allergies and systemic toxicity. These concerns have sparked interest in metal nanoparticles with antimicrobial actions [15].

Nanoparticles have a size between 1 and 100 nanometers. Their small size gives them a large surface area and enhanced chemical reactivity [16]. For antimicrobial actions, metallic nanoparticles of silver, gold, copper, iron, and zinc, with a size of 100 nm or less [17], may be suitable for antimicrobial applications [11] and for other health-related applications such as drug delivery [18].

Metal nanoparticles can interact with the EPSs of a biofilm. Metal nanoparticles are usually positively charged, while the polymer molecules of EPSs carry mostly negative charges [6]. As a result, the metal nanoparticles are attracted to the biofilm and can penetrate into the biofilm and attach to the EPSs. Once bound to bacteria or fungi, they cause disintegration of the biofilm matrix [6]. Eradication of existing biofilms will be referred to in this paper as anti-biofilm activity.

This review provides an overview of antimicrobial metallic nanoparticles and their actions on planktonic bacterial cells and on biofilms. This is followed by an analysis of recent publications and an assessment of recent patents.

## 2. Recent Advances in Nanometal Research

Key metallic nanoparticles of interest are silver, gold, copper, iron, and zinc, with the most extensive literature on silver nanoparticles as antibacterial agents.

### 2.1. Types of Metallic Nanoparticles and Their Efficacy

Nanoparticles of silver, copper, gold, zinc, titanium, and nickel exert antibacterial actions on both Gram-positive and Gram-negative bacteria [19,20,21,22,23]. They can disrupt bacterial cell walls through oxidative and nitrite stress and impair bacterial metabolism by inhibiting enzyme activity, gene expression, and ATP synthesis [15].

#### 2.1.1. Silver Nanoparticles

Silver nanoparticles (AgNPs) have remarkable antibacterial properties. They inhibit the formation of biofilms [24], including that for *Escherichia coli*, *Staphylococcus aureus*, *Enterococcus planus,* and *Pseudomonas aeruginosa* [25]. Since AgNPs penetrate into the biofilm matrix, this makes them candidates for carrying other actives (such as fungicides). They can also be used in combination with antibiotics, such as erythromycin, penicillin, or streptomycin [26]. 

Similar antibacterial activity to AgNPs is shown by silver oxide nanoparticles (AgONPs); however. a common disadvantage of metal oxide nanoparticles from silver, zinc and copper is their degradation by sunlight [27].

#### 2.1.2. Gold Nanoparticles

Gold nanoparticles (AuNPs) penetrate bacterial biofilms, impair metabolic activities of bacteria, and alter bacterial cell membranes [6]. They inhibit the production of EPSs. This effect contributes to the eradication of biofilms of *P. aeruginosa* [6], *E. coli* [28], *Pseudomonas putidis,* and *Aeromonas hydrophila*. The high price of gold makes these nanoparticles too expensive to be a first-choice antimicrobial agent [6].

#### 2.1.3. Copper Nanoparticles

Copper nanoparticles (CuNPs) are less expensive than silver or gold and have high potency as antimicrobial agents. They release positively charged copper ions (Cu^2+^) that bind to negatively charged carboxyl groups on the lipoproteins of bacterial cell membranes. The resulting increased permeability allows Cu^2+^ ions to enter bacterial cells [27], where they bind to large molecules containing sulfhydryl and phosphate residues, affecting the structure of proteins and DNA, respectively. Normal biochemical activities are impaired, leading to cell death [27]. Since copper is an essential element and is present in many enzymes in the human body, using CuNPs could destabilize normal physiological enzymatic activity. This concern limits the application of CuNPs in humans.

#### 2.1.4. Zinc Nanoparticles

Zinc oxide nanoparticles (ZnONPs) strongly inhibit the growth of both Gram-positive and Gram-negative bacteria [27], due to the release of Zn^2+^ ions. ZnONPs can prevent biofilm formation through released Zn^2+^ ions as well as by generating reactive oxygen species (ROS), which are toxic for bacteria.

#### 2.1.5. Other Metallic Nanoparticles

Other metal nanoparticles of interest include iron and manganese. These have demonstrated promising effects against *Helicobacter pylori* and against the fungal organism *Candida albicans* [15].

### 2.2. Combinations of Metal Nanoparticles with Other Agents 

Combining antibiotics with metallic nanoparticles is a topic of emerging interest. As antibiotics act on different biochemical pathways, not all antibiotics can work synergistically with metal nanoparticles. Combining AgNPs with visible blue light can enhance their antibacterial activity [29,30], and extending this to AgNPs combined with blue light and an antibiotic (amoxicillin) may further increase antimicrobial activity. This approach has been used against methicillin-resistant *Staphylococcus aureus* [29]. However, it remains to be determined whether this combination approach could still lead to the emergence of resistance to the antibiotic [31]. 

Another approach is to combine two or three antimicrobial metals together, such as silver with iron. Ag-Fe bimetallic nanoparticles interact with the thiol side chain in cysteine, changing the primary structure of proteins, increasing the permeability of bacterial cell walls, and causing bacterial death. At the same time, Ag^+^ ions released from the nanoparticles into the cytoplasm induce oxidative stress and damage DNA [32]. Synergistic effects may also occur in other bimetallic and trimetallic nanoparticles.

A summary of the literature on metal nanoparticles is provided in Table 1. This shows antibacterial and antifungal actions, as well as anti-biofilm activity (i.e., destruction of existing biofilms).

Of the various types of metal nanoparticles, AgNPs are amongst the most potent antimicrobial agents, both in terms of antibacterial actions on planktonic bacterial cultures and the prevention of biofilm formation, with the lowest MIC (0.06–20 μg/mL). Other non-silver nanoparticles are less active.

Most reported metallic nanoparticles have sizes below 100 nm, especially in the range of less than 50 nm, which is smaller than most peptides [72]. As summarized in Table 1, most metallic nanoparticles were synthesized by chemical methods, including green synthesis using plant extracts. 

### 2.3. Biofilm Inhibition by Metallic Nanoparticles

As shown in Table 1, metallic nanoparticles can impair the growth of bacteria and fungi and the formation of biofilms. They can also cause the destruction of existing biofilms, as an anti-biofilm action. These antimicrobial functions are influenced by nanoparticle type, shape, and size, since these influence their ability to create stress on cell membranes and generate ROS [73,74]. As already noted, metallic nanoparticles also release metal ions, which in turn can impair the functions of various organelles in bacteria. Mitochondrial dysfunction reduces ATP levels, while ribosome dysfunction can lead to enzyme inactivation and protein denaturation (Figure 2) [75].

### 2.4. Methods for the Synthesis of Silver Nanoparticles

Since AgNPs are currently the most extensively explored type of metal nanoparticle for treating biofilms or preventing their formation, they serve as an exemplar for other metallic nanoparticles. Such particles can be synthesized or fabricated using a range of approaches, which can be divided into “top-down” or “bottom-up” (Figure 3) [76]. In the former method, large particles are reduced in size to particles in the nanoscale range. The second method, also known as the self-assembly method, involves forming nanoparticles from atoms or ions of the metal [77,78]. This can be conducted in a chemical reaction in a laboratory vessel or inside a plant or microorganism. Methods using extracts of plants are becoming popular for “green synthesis” since they may use waste products.

#### Methods for Synthesis of AgNPs

A.Chemical methods

In bottom-up chemical synthesis methods, precursors (e.g., silver ions) are transformed using reducing agents [79]. The precursor material (usually silver nitrate) is relatively expensive. The characteristics of the AgNPs that are produced vary according to the starting materials that are selected (precursor, reducing agent, solvent, and stabilizer or end-sealer) [76] and their concentrations, as well as the temperature [80]. Stabilizers or capping agents are usually included to ensure the stability of the AgNPs. Attention must also be paid to by-products that are generated by the synthesis reaction [81]. These issues have driven interest in synthesis methods that are simpler and less expensive and that are still capable of producing AgNPs of a suitable size and that have high stability [82].

B.Physical methods

These are top-down methods and include using heat [83], plasma [84], electromagnetic radiation [85], arc discharge, and lithography [86]. Physical methods are considered labor-intensive and costly.

C.Biological methods including green synthesis

As shown in Table 1, a range of methods using plant extracts and microorganisms have been used to convert silver ions into AgNPs. These methods do not require complex equipment and are considered to be more environmentally friendly and lower in cost than physical or chemical methods [87,88,89]. The use of green synthesis methods to prepare metal nanoparticles is an emerging trend in nanotechnology [90].

### 2.5. Biosafety of AgNPs When Applied onto the Human Body

Unlike zinc, iron, copper, and cobalt, which are essential trace elements in the human body, silver is not an essential component of the human diet and does not play a role in normal human physiological processes. Ingestion of modest amounts of ionic silver can lead to silver accumulation in the human body. For example, a dose of 4.1 g of silver arsphenamine (0.6 g silver) can cause visible discoloration of the skin, known as argyria [91], due to accumulation of silver in the skin [92]. Once ingested, silver may also reach tissues other than the skin [93], including the kidneys and the respiratory system [93], the gastrointestinal system [93,94], the female genitourinary system [95], and the brain [94]. 

AgNPs applied topically to the skin may penetrate through normal intact tissues. For example, AgNPs smaller than 10 nanometers can pass through the pores of the stratum corneum easily, while AgNPs around 7–20 nm in size can only penetrate sweat glands and at hair follicle sebaceous glands. AgNPs of 20–200 nm tend to stay at the opening of hair follicles and tend not to penetrate the skin [93]. 

Adverse effects from prolonged exposure to silver are dose-related [94]. Ingestion of AgNPs and ionic silver can lead to liver and kidney damage. The toxicity of AgNPs is influenced by particle size, particle shape, and the presence of a capping or end-sealing agent [92]. Oral toxicity tests using adult male rats have shown that ingested AgNPs can alter the gut microbiome [96], which could have downstream health impacts. 

Retained silver in the human body following its ingestion can cause issues, with a notable example being from particles in renal glomeruli [97]. The most clinically noticeable effects are those seen in the skin (Figure 4); however, the accumulation of AgNPs in anyone site of the body is a concern in terms of toxicity [98]. Despite occasional reports of argyria, most studies have concluded that silver does not cause significant injuries that could lead to death [92]. 

### 2.6. Analysis of the Literature from 2004 to 2024

An analysis of 20 years of the literature on metallic nanoparticles for antimicrobial treatment was undertaken following the PRISMA model. Searches were made on ScienceDirect, Scopus, and Web of Science databases (Figure 5). After removing duplicates and irrelevant or incomplete reports, this led to 35 publications over the period which were highly related to the topic. 

Several trends in publications on metallic nanoparticles are evident over 20 years. There was a steady increase each year in papers on monometallic and bimetallic nanoparticles, with more sharply rising interest in the latter from 2019 onwards (Figure 6). 

A patent search for metallic nanoparticles used for anti-biofilm actions was made using the Espacenet patent database for the period of 2003 to 2023 (Figure 7). The number of patent family applications per year rose over the period, with the cumulative count reaching over 3000 by the year 2023. This shows sustained activity in developing and protecting intellectual property around the use of metallic nanoparticles as antimicrobial agents. 

An analysis of the inventor country or origin for patent families is shown in Figure 8. Major areas were the US, the EU, and China. A large number of patents (over 2000) had reached the international phase (WO/World Intellectual Property Organization). 

Figure 9 presents a summary of patent applications (“reports”) and granted patents over the 20-year period of interest, for metallic nanoparticles used for antimicrobial treatments. This shows strong growth in patent applications, with a relatively steady number of granted patents each year over the most recent decade. 

A further approach was to use VOS Viewer network visualization software (version 1.6.20) to identify the bibliometric network for the 35 included papers from the PRISMA screening process. The first analysis was for silver nanoparticles (Figure 10). Two colors were used to denote clusters. Red represents key words around AgNP properties, such as physicochemical characterization and synthesis, while blue represents performance aspects relevant to safety and effectiveness, such as antimicrobial potency, cytotoxicity, and metabolism. 

A parallel visualization was prepared for zinc-containing nanoparticles (Figure 11), following the same color coding.

The third visualization was prepared for copper-containing nanoparticles (Figure 12).

The final visualization was prepared for gold nanoparticles (Figure 13).

Looking across the networks, the busiest networking visualization is for silver-containing nanoparticles, which reflects the more extensive literature on this topic when compared to other metallic nanoparticles. 

A common theme across all the network analyses was that assessments of the antibacterial activity of metal nanoparticles were mostly confined to in vitro studies, especially using single-species biofilm models. This reinforces the need for more complex assessments including complex biofilms in the laboratory, followed by appropriate animal studies and human clinical trials. 

## 3. Conclusions and Future Directions

Potential candidates for eradicating biofilms, and preventing their development, include nanoparticles of silver, gold, copper, iron, manganese, nickel, and zinc, and their oxides, used singly or in various combinations. Of these, AgNPs have been the most extensively investigated and appear to have the greatest potency as anti-biofilm agents. Metal nanoparticles are a promising way to prevent the growth of bacteria and fungi, and their formation of biofilms. In addition, metal nanoparticles exhibit useful anti-biofilm actions that lead to the eradication of established biofilms. 

More attention should be directed to exploring how well metal nanoparticles penetrate into biofilms of different types and whether they provide any long-term influence on biofilm characteristics, such as through the slow generation of metal ions over long periods of time. Likewise, more exploration into how metal nanoparticles interact with quorum-sensing mechanisms is needed.

Multiple methods exist for synthesizing metal nanoparticles, and the choice of synthesis parameters determined key properties of nanoparticles such as their size and shape. Green synthesis methods offer low cost and simplicity and are used widely. Nanoparticle characteristics can be tuned to optimize antimicrobial actions, whilst maximizing biocompatibility and minimizing toxicity.

Within a biofilm, metal nanoparticles can interact with microorganisms as well as with EPSs. It is possible to combine metal nanoparticles with one another, or with antibiotics, to increase their effects on established biofilms. A review of two decades of literature showed growing interest in bimetallic nanoparticles.

The question of whether metal nanoparticles avoid altogether the problems of resistance requires further study, especially since some nanoparticles are known to bind to bacterial DNA [6], as well as to bacterial cell membranes. The extent to which the same nanoparticles can enter human cells and influence human DNA also requires close examination [15].

A network analysis of publications showed that most research was conducted in laboratory settings using single-species biofilms. The actions of metal nanoparticles as anti-biofilm agents must be tested with more complex biofilms in the laboratory setting and then in animal models and in human clinical trials. In the latter, issues with AgNPs could arise if these are in prolonged contact for sufficient time for the particles to penetrate into the body. Hence, there is a need for careful evaluation of their effects on human health.

The analysis of patent applications and granted patents shows strong interest internationally in extending knowledge regarding metal nanoparticles as antimicrobial agents. There is strong interest in targeted therapies that combine metal nanoparticles with antibiotics to combat drug-resistant infections [92]. This drive to enhance effectiveness must be accompanied by careful exploration of safety aspects, especially mucosal surface absorption and the accumulation of metal nanoparticles in organs, as part of the studies of the safety of using metal nanoparticles as therapeutic agents [15]. An emphasis on establishing comprehensive data on safety aspects is essential before considering their clinical application in human health care. 

## Figures and Tables

**Figure 1 antibiotics-13-00819-f001:**
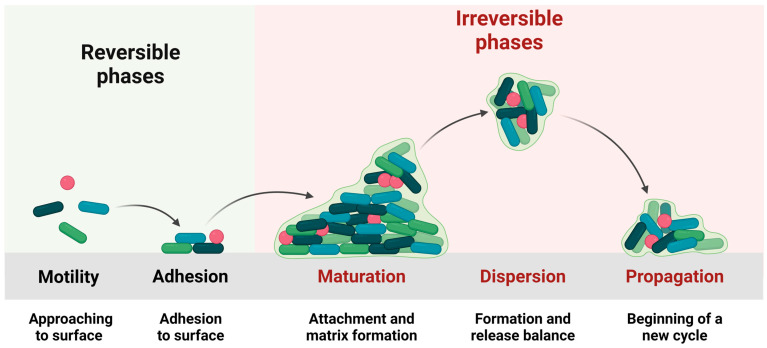
The main stages of biofilm formation. Figure created using ©2024 Biorender.

**Figure 2 antibiotics-13-00819-f002:**
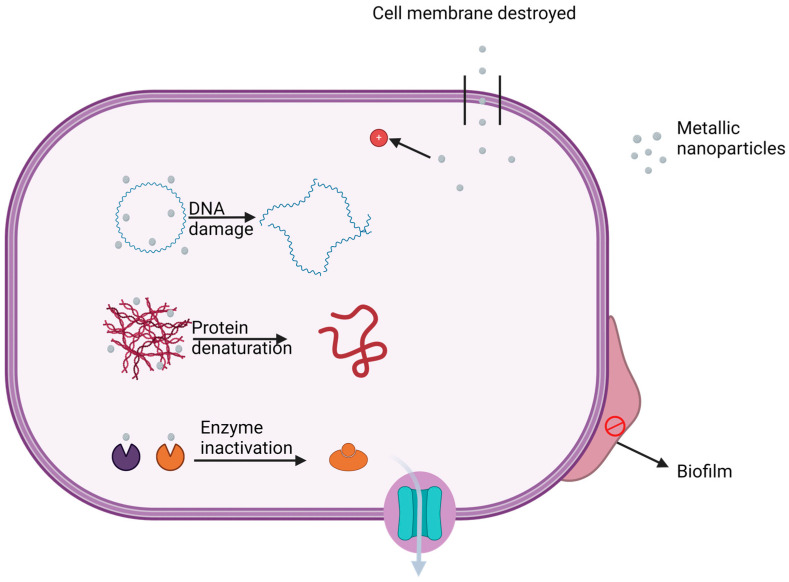
Selected antibacterial actions of metallic nanoparticles. Image created using ©2024 Biorender.

**Figure 3 antibiotics-13-00819-f003:**
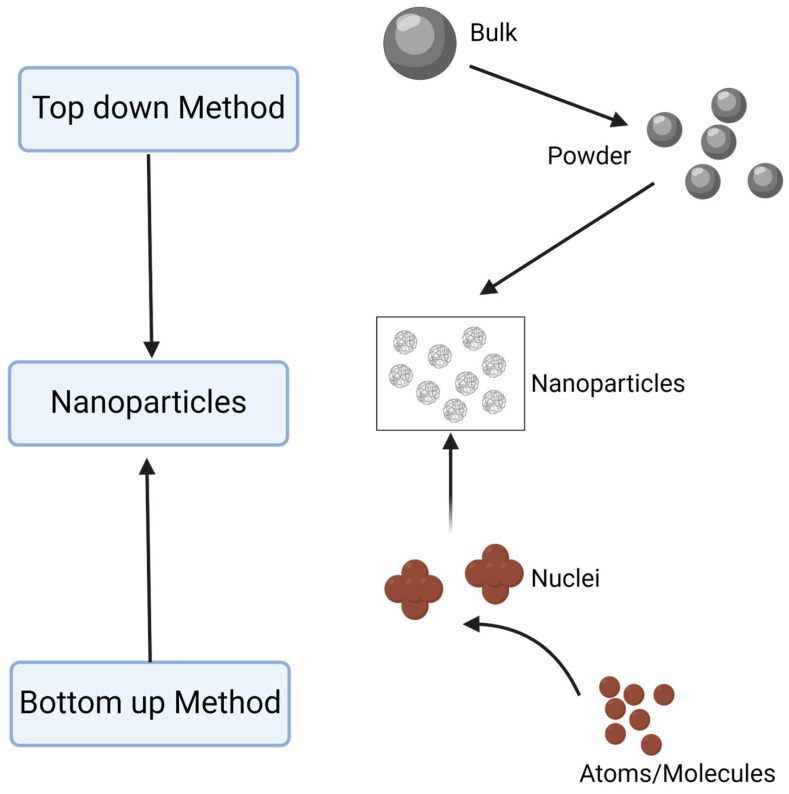
Two main strategies for the synthesis of metallic nanoparticles. Biological methods such as green synthesis follow the bottom-up approach and reduce metal ions to metal nanoparticles. Image created using ©2024 Biorender.

**Figure 4 antibiotics-13-00819-f004:**
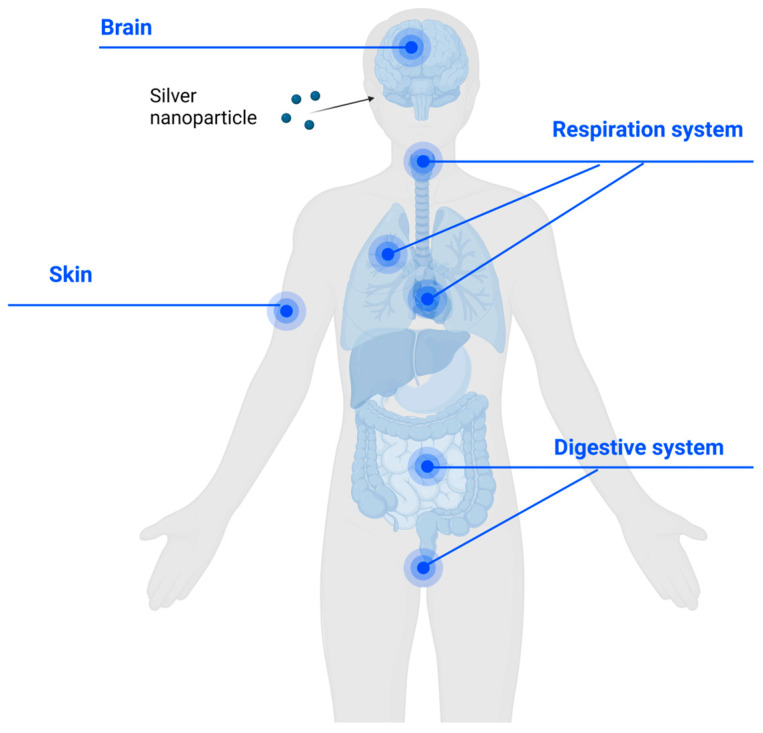
Entry of AgNPs into the human body (by ingestion, inhalation, and topical application) and some major sites where silver can accumulate. Image created using ©2024 Biorender.

**Figure 5 antibiotics-13-00819-f005:**
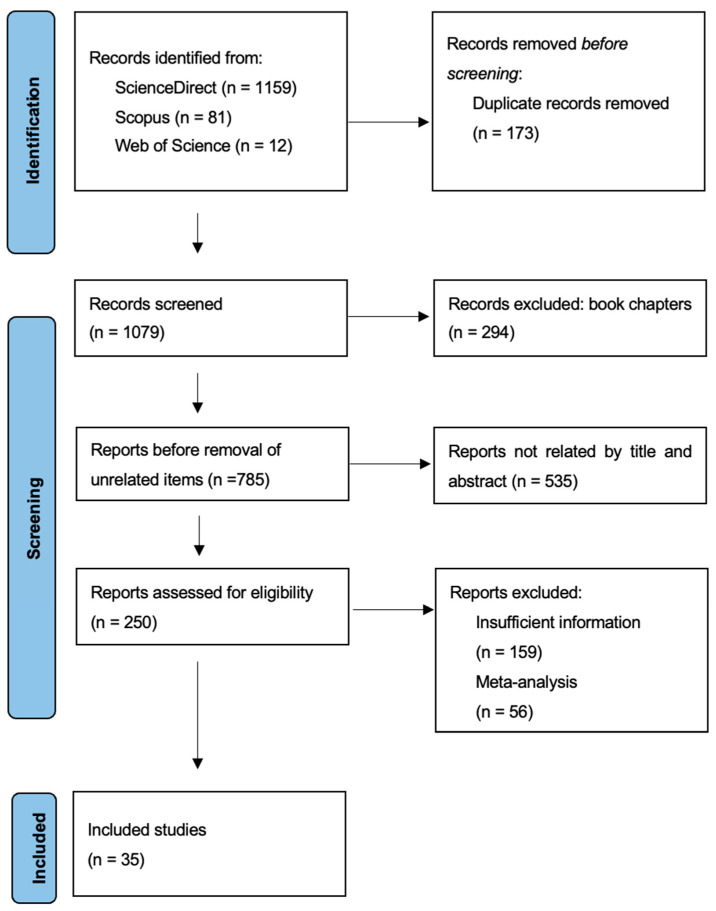
PRISMA flow diagram for the included studies from 2004 to 2024.

**Figure 6 antibiotics-13-00819-f006:**
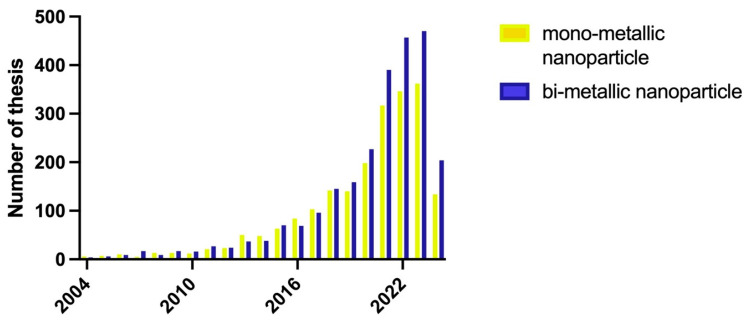
Publications per year on single metallic nanoparticles and bimetallic nanoparticles over 2004–2024. This graph was prepared using Prism 10 software. Note that for 2024, only the period from January to June 2024 is included.

**Figure 7 antibiotics-13-00819-f007:**
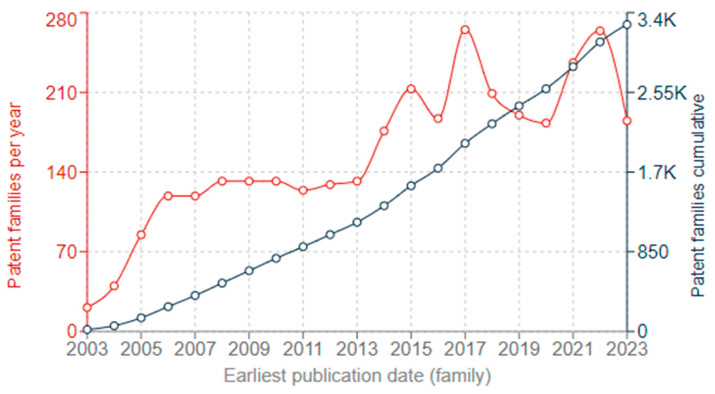
The yearly and cumulative patent trend in last two decades. Data is obtained from searching the Espacenet database.

**Figure 8 antibiotics-13-00819-f008:**
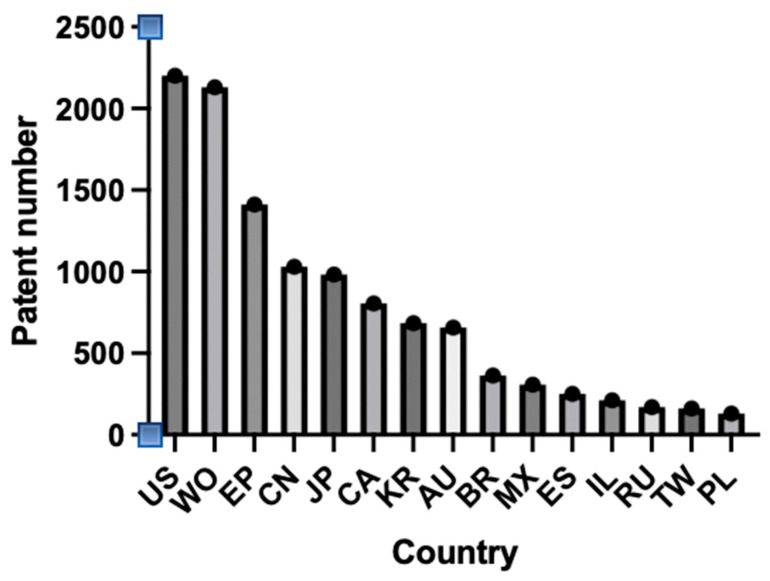
Metallic nanoparticle patent numbers by jurisdiction. This figure was prepared using Prism 10 software.

**Figure 9 antibiotics-13-00819-f009:**
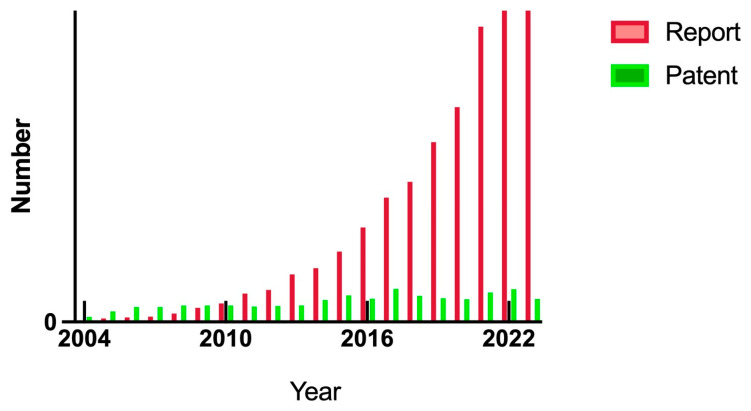
Patent applications (“reports”) and granted patents per year from 2004 to 2024 for metallic nanoparticles used for antimicrobial treatments. This figure was prepared using Prism 10 software.

**Figure 10 antibiotics-13-00819-f010:**
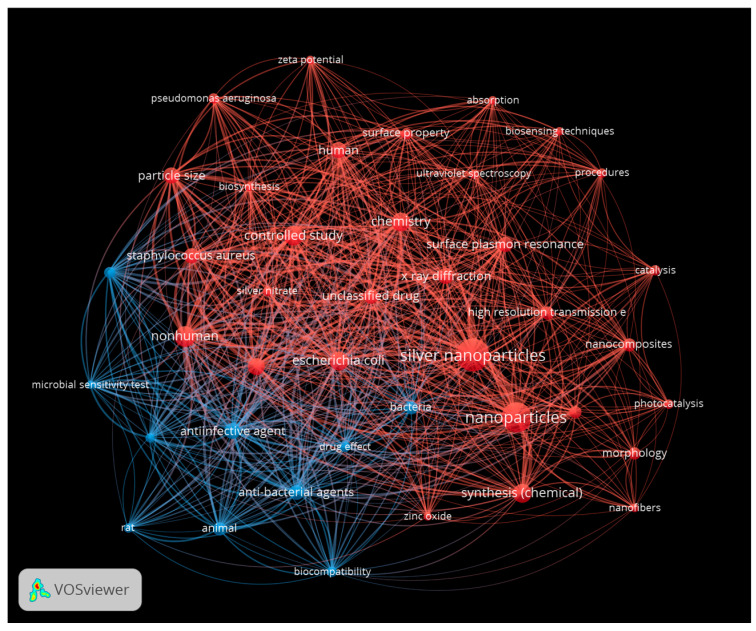
Network analysis for silver nanoparticles, showing nodes for nanoparticle characterization (red) and for antimicrobial potency aspects (blue). This figure was prepared using VOS Viewer.

**Figure 11 antibiotics-13-00819-f011:**
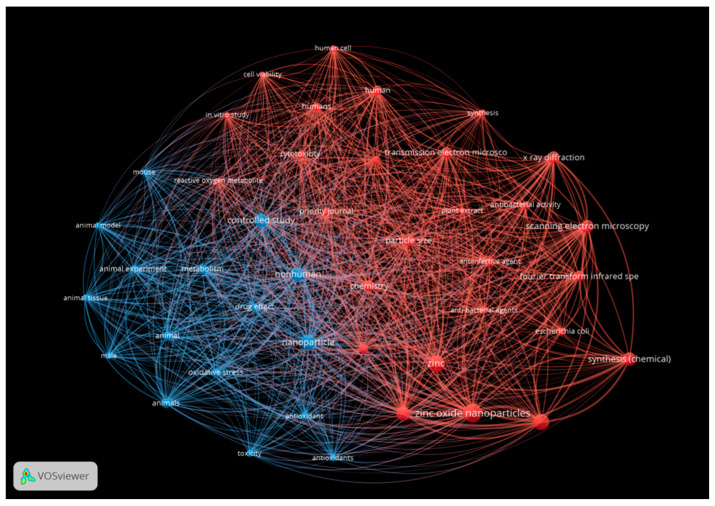
Network analysis for zinc nanoparticles showing characterization (red) and performance (blue). This figure was prepared using VOS Viewer software.

**Figure 12 antibiotics-13-00819-f012:**
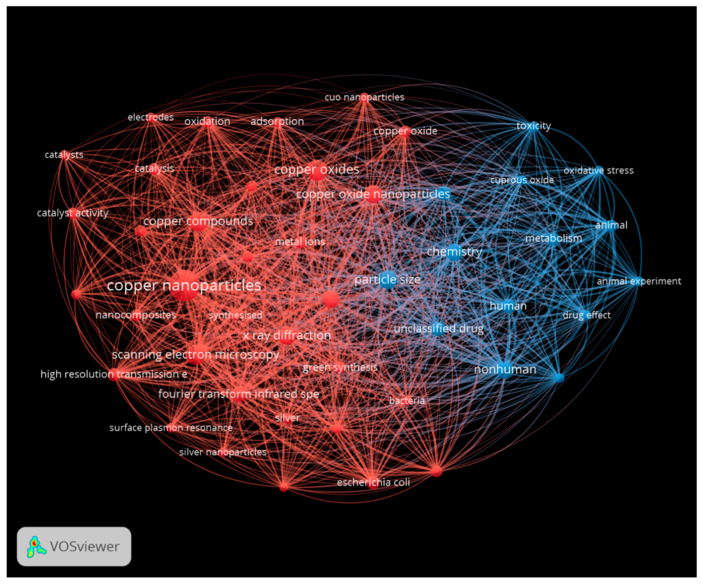
Network analysis for copper nanoparticles, showing characterization (red) and performance (blue). This figure was prepared using VOS Viewer software.

**Figure 13 antibiotics-13-00819-f013:**
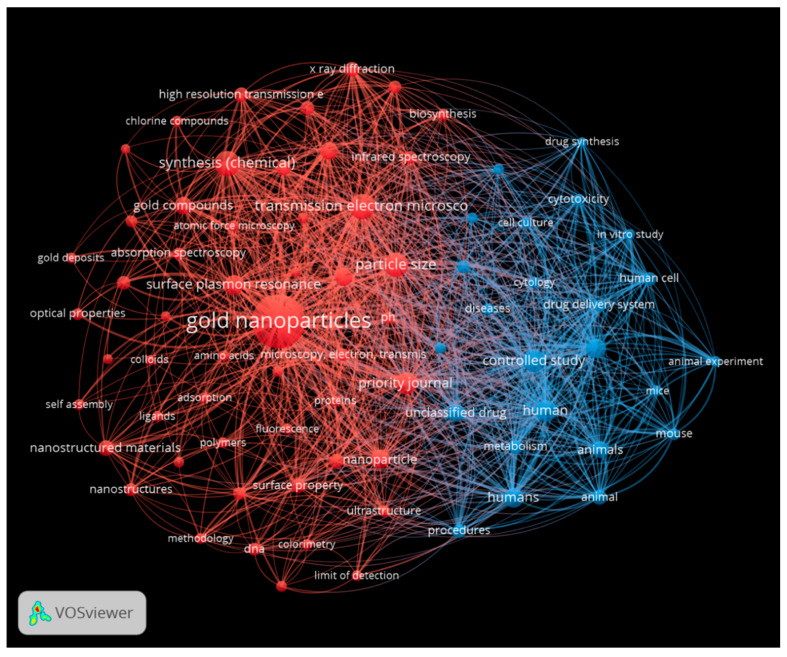
Network analysis for gold nanoparticles, showing characterization (red) and performance (blue). This figure was prepared using VOS Viewer software.

**Table 1 antibiotics-13-00819-t001:** Metallic nanoparticles as antimicrobial agents used alone and in complexes to treat planktonic bacteria and to reduce or eradicate existing biofilms.

Metallic Nanoparticles (Reference)		Size (nm) and Shape	Synthesis Resource	MIC (μg/mL) Reduction Ratio Inhibition Zone (mm)	Microorganism	Comments
Silver	Silver (AgNPs) [33]	50 Hexagonal	AgNO_3_ + leaves of Geranium	0.18 - 8.5	*P. aeruginosa* ATCC-27853	Antibacterial actions
	Silver (AgNPs) [34]	100 Spherical	AgNO_3_ + malt extract	3.75–15 60–100% -	*S. aureus* ATCC 25904	For standard strains and MDR isolates and reduced and eradicated biofilms. Anti-biofilm actions.
	Silver (AgNPs) [35]	1.2–62 Spherical	AgNO_3_ + *L. acapulcensis* extract	0.06 99.9% 16	*S. aureus* ATCC 49476	Significant antimicrobial effects on *S. aureus* and *P. aeruginosa*
				0.06 99.9% 15	*P. aeruginosa* *ATCC 27853*	
	Silver (AgNPs) [36]	32.2 Spherical	AgNO_3_ + starch broth medium	- 10.7% 34.6% 39.08% 34.75% -	*P. aeruginosa* ATCC 9027	Antibacterial and anti-biofilm actions
					*S. typhi* ATCC 12023	
					*S. aureus* ATCC 6598	
					*E. coli* ATCC 8739	
Silver oxide	Silver oxide (AgONPs) [37]	10–25 Spherical	Fresh *Telfairia occidentalis leaves* + AgNO_3_	20 - 15	*K. pneumoniae * isolated from humans	Strong antibacterial activity
	Silver oxide (AgONPs) [38]	10–50 Spherical	*Actinomycetes* spp. + AgNO_3_	500 - 15	*MDR P. aeruginosa*	Antibacterial activity
Silver-containing nanoparticles	Ag–TiO_2_ NPs [39]	31.3 ± 0.5 or 23.4 ± 0.4 -	TiO_2_ nanopowder	150 - -	*B. subtilis* IC 12488	Antibacterial and anti-biofilm activity
				150 -	*S. aureus* IC 13204	
			AgNO_3_	1200 -	*E. coli* IC 13529	
				1200 -	*K. pneumoniae* IC 13420	
			PSSA-co-MA	1200 -	*P. aeruginosa* IC 13202	
	Ag-Fe NPs [32]	13 -	*G*. *jasminoides*	65 - -	*C. albicans*	Growth inhibition
			AgNO_3_			
			Fe(NO_3_)_3_			
	Ag-Ni nanoparticles [40]	31.84–47.85 -	AgNO_3_	1.56 *-* -	*C. albicans* SC 5314	Growth inhibition and anti-biofilm activity
			Ni(NO_3_)_2_·6H_2_O			Inhibition of morphogenesis
			*Salvia officinalis*			Reduced efflux pump genes
						ROS production
	Fluorinated Phthalocyanine–silver NPs [41]	15–20 -	Silver nanoparticles	8 95–100% -	*P. aeruginosa* ATCC 27853	Antioxidant effects
			Phthalonitrile	8 99–100% -	*E. coli* ATCC 25922	Antimicrobial and anti-biofilm actions
			Phthalocyanines			Antimicrobial photodynamic therapy activities
	Ag-Se nanoparticles [42]	66.5 -	AgNO_3_	62.5 90.7% -	*C. albicans* ATCC 10231	Antifungal
			Sodium selenite	62.5 90.6% -		Antibacterial activity
					*E. coli* ATCC 11229	Anti-biofilm actions
				125 86.2% -	*P. aeruginosa* ATCC 6538	Free radical scavenging
	Silver–curcumin NPs [43]	30 -	AgNO_3_	3 - -	*P. aeruginosa* PA01	Antibacterial and anti-biofilm actions
			Gallic acid			
			Curcumin	50 - -	*S. aureus* ATCC 25923	
			PVP			
	Ag-ZnO NPs [44]	28 -	*E. scaber* leaves	0.125 62.5% -	*S. aureus*	Antibacterial actions
			Zinc nitrate		*B. subtilis*	Anti-biofilm actions
			AgNO_3_			Antioxidant
Gold	Gold (AuNPs) [45]	5–50 Spherical	Trisodium citrate HAuCl_4_·4H_2_O	18.71 80.4% 6.3	*S. marcescens* Clinical collection	Antibacterial and anti-biofilm activity, through production of ROS
	Gold(AuNPs) [46]	20–100 Hexagonal	Marine alga G. elongate HAuCl_4_ aqueous solution	- - 13–17	*K. pneumoniae* ATCC 27738	Antibacterial activity
	Gold(AuNPs) [47]	18 Spherical	HAuCl_4_·4H_2_O Trisodium citrate solution	- - -	*S. aureus * collected from volunteer	/
	Gold(AuNPs) [48]	11 ± 3 Spherical	Cinnamaldehyde HAuCl_4_·4H_2_O	- 93% -	*C. albicans* DAY185	Anti-biofilm actions
	Gold(AuNPs) [49]	43 Spherical	Chloroauric acid Citrate	120 - -	*E. coli * ATCC25992	Antibacterial and anti-biofilm actions
					*S. epidermidis* 1301	
				1 - 15.5	*C. krusei* not determined	
				1 - 20.5	*C. glabrata* not determined	
Copper	Copper (CuNPs) [50]	28.3 (C), 43.8 (M) Spherical	*Cassia fistula* (*C*) and *Melia azedarach* (*M*) leaves + cupric nitrate trihydrate	1000 99.8%,92.5% 15.13	*K. pneumonia * from collection center	Antibacterial and anti-biofilm actions toward both species
				1000 100%,99.5% 2.6	*H. pylori * from collection center	
Copper oxide	Copper oxide (CuONPs) [51]	80–300 Spherical	Cu_2_O microparticles + ethanol	- 98% -	*MDR E. coli*	Antibacterial and anti-biofilm actions
				- 99.2% -	*MDR S. aureus*	
	Copper oxide (CuONPs) [52]	10–12 Spherical	Copper sulfate *T. chebula* extract	1000 – 15.67	*E. coli * not determined	Antibacterial and anti-biofilm actions
				1000 - 16	*S. aureus* not determined	
				750 - 17	*P. aeruginosa* not determined	
	Copper oxide(CuONPs) [53]	28–33 Spherical	Copper sulfate solution Leaf extract	125 - 12–13	*A. baumannii* MH 605335	Antibacterial and anti-biofilm actions
	Copper oxide(CuONPs) [54]	20 Spherical	Copper sulfate (CuSO_4_∙5H_2_O) Cell free supernatant	1000 96% -	*E. coli * from mat	Anti-biofilm actions
Zinc oxide	Zinc oxide (ZnONPs) [55]	40–130 Spikes	Zinc acetate Dihydrate DDW CTAB	800 80–85% -	*P. aeruginosa* PAO 1	Anti-biofilm actions and inhibition of QS for resistant *P. aeruginosa*
				500 80% -	*C. violaceum* CVO26	
	Zinc oxide–Xanthan nanocomposite [56]	16 Rod	Zn(NO_3_)_2_∙6H_2_O + xanthan gum + NaOH	256 70% -	*C. violaceum* ATCC 12472	Anti-biofilm actions
				256 75% -	*S. marcescens* ATCC 13880	
	Zinc oxide (ZnONPs) [57]	4.4 Spherical hexagonal	Zn(Ac)_2_∙2H_2_O Anhydrous methanol KOH	500 98.5% -	*S. aureus* SH 1000	Antibacterial and anti-biofilm actions
				500 - -	*E. coli* UT 189	
	Zinc oxide (ZnONPs) [58]	24.62 Spherical spheroidal	*Plumbago zeylanica* L. Zinc acetate	- 52.69% -	*E. coli* ATCC 25922	Inhibition of biofilm growth and anti-biofilm actions
				- 59.79% -	*S. aureus* MTCC 3160	
				- 67.22% -	*P. aeruginosa* PAO 1	
Zinc-containing nanoparticles	Zn-doped CuO NPs [59]	- -	Copper zinc acetates	- 91% -	*E. coli* ATCC 25922	Inhibition of biofilm formation and anti-biofilm actions
			Aqueous ammonium hydroxide	- 92% -	*S. aureus* ATCC 29213	
			Ethanol	- 95% -	*P. mirabilis* not determined	
	Curcumin-ZnO NPs [60]	110.51 -	Zinc nitrate hexahydrate	62.5 45–90% -	*P. aeruginosa* PAO 1	Anti-biofilm actions
			2-Thiobarbituric acid			
	ZnMgO NPs [61]	10 -	MgO	- 61% -	*E. coli* BL21 DE3	Antibacterial actions
			ZnO	- 25% -	*B. subtilis* 168	
	ZnCuFe NPs [62]	42 -	Zinc acetate dihydrate	150 85% -	*E. coli * from chronic infection	Antibacterial and anti-biofilm activity
			Copper acetate hydrate			
			Iron nitrate nonahydrate	150 55% -	*E. faecalis * from chronic infection	
			n-propyl amine			
	ZnO-Au hybrid NPs [63]	30 -	ZnO	- 90% -	*S. aureus* not determined	Antibacterial effects
			AuCl_4_^−^		*E. coli* not determined	
Cobalt-containing nanoparticles	CoFe_2_O_4_ NPs [64]	10 -	Iron nitrate	5000 - -	*C. albicans* ATCC 10231	Antimicrobial effects and anti-biofilm actions
			Cobalt nitrate	5000 - -	*P. aeruginosa* ATCC 27853	
			Eucalyptus plant extract	5000 - -	*E. coli* ATCC 25922	
Iron oxide	Iron oxide NPs [65]	10–11 Spherical	Ferrous chloride tetrahydrate (FeCl_2_∙4H_2_O)	50 - -	*E. coli* not determined	Antimicrobial effects and anti-biofilm actions
			Ferric chloride hexahydrate (FeCl_3_∙6H_2_O)			
Titanium Dioxide	TiO_2_ NPs [66]	39.2 Rods	Titanium rods, Chitosan, Alginic acid sodium salt	400 - -	*S. aureus* DNC274 ATCC 29213	Antibacterial and anti-biofilm actions
Selenium	Se NPs [67]	23.47 Red spherical	Sodium selenite DPPH Sodium borohydride	25 - -	*C. albicans * IFRC 1873	Antifungal activity against tested fungi strains
					*F. proliferatum* IFRC 1871	
					*F. equiseti* IFRC 1872	
					*T. mentagrophytes* FR5_22130	
					*A. fumigatus* IFRC 1649	
Magnesium oxide	MgO NPs [68]	50–70 spherical	NaOH, NaNO_3_, MgCl_2_	250 82.9% 28 ± 0.33 mm (2000 μg/mL) 21 ± 0.288 mm (1000 μg/mL) 17.5 ± 0.288 mm (500 μg/mL)	*E. coli* KT273995	Antibacterial activity and anti-biofilm actions
				125 82.9% 35.5 ± 0.33 mm (2000 μg/mL) 35.5 ± 0.288 mm (1000 μg/mL) 30.5 ± 1.90 mm (500 μg/mL)	*Klebsiella pneumoniae* KT273996	
				500 82.9% 25.5 ± 0.268 mm (2000 μg/mL) 23.5 ± 1.32 mm (1000 μg/mL) 20.5 ± 2.18 mm (500 μg/mL)	*Staphylococcus aureus* KT250728	
	[69]	50–100 irregular but spherical particle-like shapes	Purchased from Sigma Aldrich Chemical Co. (Saint Louis, MO, USA)	200 93.40 to 95.60% 14.30 mm	*R. solanacearum * from infected tobacco	Bacteriostatic at low concentrations and anti-biofilm activity
	[70]	4 square and polyhedral shape	Mg(CH_3_COO)_2_, Aerogel AP-MgO	625 99.9% -	*E. coli* not determined	Antibacterial effects and anti-biofilm activity
				625 95% -	*S. aureus* not determined	
Nickle oxide	NiO NPs [71]	14 ± 5.8 polymorphic	Eucalyptus leaf extract (ELE), Ni(NO_3_)_2_·6H_2_O	800 - 15	*Methicillin sensitive S. aureus*-06	Antibacterial and anti-biofilm activity against the tested strains
				800 - 13	*Methicillin sensitive S. aureus*-02	
				800 - 15	*P. aeruginosa*-48	
				800 - 14	*P. aeruginosa*-64	
				1600 - 17	*E. coli*-60	
				800 - 17	*E. coli*-52	
				800 - 15	*Methicillin-resistant S. aureus*-10	
				800 - 14	*Methicillin-resistant S. aureus*-31	

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
