# Peer review of "Characteristics of Metallic Nanoparticles (Especially Silver Nanoparticles) as Anti-Biofilm Agents"

_antibiotics, 2024, doi:10.3390/antibiotics13090819_

Round 1

Reviewer 1 Report

Comments and Suggestions for Authors

The author reviewed the the metallic nanoparticles, compared their role for treating planktonic and biofilm in this manuscript. As a review to antibiotics, the author tried to include most relevant contents in the manuscript. The organization of the article is similar to that of a general review type. But there are still the following problems need to modify:

1. For the title, it seems to be too big, the author can try to write more close to the content of the manuscript.

2. For the keywords, the authors need to further optimize.

3. The author introduced many kinds of metal particles in the manuscript, but it seems to introduce silver ions in more detail. This made the structure of the article a little strange. Why do you do this?

4. The quality of the pictures in the article needs to be improved.

5. As for the conclusion part, the author does not give many research suggestions and prospects, which must be strengthened.

6. For references, the author needs to check again and be sure to include all relevant references in recent years. In addition, according to the content of this review, the author's references appear to be less.

7. For a review type, the author should choose to focus on writing some content and delete some unnecessary content.

8. It is recommended that the author use more pictures to illustrate the relevant important content of the article, so that the reader can easily understand.

Author Response

Reviewer 1

The author reviewed the metallic nanoparticles, compared their role for treating planktonic and biofilm in this manuscript. As a review to antibiotics, the author tried to include most relevant contents in the manuscript. The organization of the article is similar to that of a general review type. But there are still the following problems need to modify:

  1. For the title, it seems to be too big, the author can try to write more close to the content of the manuscript.

Response: As suggested, we narrowed the title to now reflect the focus of the paper.

  1. For the keywords, the authors need to further optimize.

Response: We changed the keywords to “Antimicrobial Agents, Biofilm, Metallic nanoparticles, Bimetallic nanoparticles, Silver nanoparticles”

  1. The authors introduced many kinds of metal particles in the manuscript, but it seems to introduce silver ions in more detail. This made the structure of the article a little strange. Why do you do this?

Response: Our intention was to introduce the topic of metallic NPs as antimicrobials, and then focus on silver nanoparticles, as these are the type most investigated and reported in the literature, and they are exemplars for others. We now explain that in the text. This is also why the title and abstract have been changed to reflect that emphasis.

  1. The quality of the pictures in the article needs to be improved.

Response: We have revised all the figures to enhance their quality.

  1. As for the conclusion part, the author does not give many research suggestions and prospects, which must be strengthened.

Response: We rewrote the entire conclusions section, and added further text with suggestions for future work.

  1. For references, the author needs to check again and be sure to include all relevant references in recent years. In addition, according to the content of this review, the author's references appear to be less.

Response: We undertook an analysis of relevant publications from 2004-2024. In so doing, we included relevant recent papers from the literature that directly related to the topic.

  1. For a review type, the author should choose to focus on writing some content and delete some unnecessary content.

Response: We went through the entire paper and reduced the volume of content to increase the focus of the article on using metallic nanoparticles to treat biofilm.

  1. It is recommended that the author use more pictures to illustrate the relevant important content of the article, so that the reader can easily understand.

Response: The revised paper now has a total of 13 figures that cover all the key points raised in the text.

Reviewer 2 Report

Comments and Suggestions for Authors

I appreciate the authors' contribution to the review on "Metallic Nanoparticles as Anti-Biofilm Agents" presenting a detail about the types, synthesis methods, and application of different nanoparticles as anti-Biofilm agents. The topic is interesting; however, the authors present the review articles superficially rather than focusing deeply on the subject matter. Therefore, I recommend a major revision for this manuscript and suggest overcoming the attached issues.

Comments on the Quality of English Language

There are many grammar, spelling, comma, and space mistakes throughout the manuscript.  English editing is required throughout the manuscript.  

Author Response

Reviewer 2

I appreciate the authors' contribution to the review on "Metallic Nanoparticles as Anti-Biofilm Agents" presenting a detail about the types, synthesis methods, and application of different nanoparticles as anti-Biofilm agents. The topic is interesting; however, the authors present the review articles superficially rather than focusing deep into the subject matter.

Therefore, I recommend a major revision for this manuscript and suggest overcoming the following issues.

  1. There are so many mistakes in the grammar, spelling, comma, and space throughout the manuscript. English editing is required throughout the manuscript.

Response: We have undertaken a detailed review of the whole manuscript to address errors in the text and the table.

  1. The authors should present a clear flow chart or graphical representation of their main idea of the review article in a single frame.

Response: We created a graphical abstract which captures this, and it appears at the start of the paper.

  1. The authors describe the introduction in sub heading titles. The author could remove the sub heading and concise the introduction, emphasizing on abstract and title. The short paragraphs could be merged (line 118-120, 121-128). In addition, a separate main title could be used for the section of types of biofilm formation, interaction between different species of biofilms and metallic interactions.

Response: In line with these suggestions, the subheadings have been changed and the naming and numbering of sections has been changed (Pages 2-4).

  1. It would be better to elaborate what are the traditional bacterial biofilm suppression methods through table. There is lack of citation of traditional bacterial biofilm suppression methods.

Response: There are many traditional methods, none of which relate to antimicrobial metallic nanoparticles, which is why they are not included in Table 1. However, a brief mention of these and their limitations has been made in the main text.

  1. Suggestion for revising the Table 1. It would be better to write the name of metallic nanoparticles in horizontal way and reference no. in separate column. Why biofilm and anti-biofilm are bold?

Response: For efficient use of space, we have reorganized Table 1 so the particle type and references are aligned vertically. We took off the bold font. 

  1. Check the lower subscript, charge, and molecular structures of the nanoparticles (structures of citations [33], [41], [57], [64], [66] in the table). S.aureus and P.aeruginosa should be italic [comments of silver nanoparticle [36]. Please cite the references of the table to text body too.

Response: We edited the table in line with these points, included citations, and made species names italicized.

  1. Line 256, mention table no. Check the silver spelling. Line 257-264, add references.

Response. We cited the table, corrected the spelling, and added in a reference.

  1. After section 2.3. “Bioprocesses in biofilm inhibition by metallic nanoparticles” [Line 265-286], suddenly the topic “General methodology for synthesis of silver nanoparticles” [Line 287] without section no. Also, there is no relation between section 2.3. and subsection 2.3.1.

Response: We have modified the text, and what was section 2.3.1 is now section 2.4. Hence 2.3 is “Biofilm inhibition by metallic nanoparticles”, while the following section 2.4 is about the synthesis of silver nanoparticles.

  1. Authors mentioned the several methods for synthesis of AgNO3 [Line 306]. It would be better to mention the sub section as a, b and c for methods or spacing in between the paragraphs.

Response: On pages 12, we made these into sequential separate spaced paragraphs A, B and C, as suggested.

  1. The main point is table 2 and 3 are missing in the text. The citation style is different in line no. 319. Biological method for the synthesis of AgNO3 is should be illustrated in the figure 3.

Response: We corrected the table numbering. Now there is only one table in the text (Table 1). We fixed the issue with the citation that was identified. Figure 3 shows the two main approaches to metallic nanoparticle synthesis, and the legend refers to biological methods.

  1. In section 2.4 (line 328-354), authors discussed the toxicity or biosafety of AgNO3. Authors could highlight a range of size of AgNO3 that have been proved to be non-toxic to human and provide additional citations to support these non-toxicity limits. Figure 4 appears blurry and should be replaced with clear version. Revise the figure legend.

Response: This section of the text now gives the fate of silver nanoparticles of different sizes (smaller than 10 nm, 7-20 nm, and 20-200 nm. Additional text has also been added on silver toxicity. Figure 4 has been redone at high resolution.

  1. There is a good point that authors demonstrated the meta-analysis of article.

Thank you for this supportive comment.

  1. Figure 5 should be re-arranged or re-made clearly. Please check the arrow alignments. No. of publication after removable replications is vary in text (n = 1078) and figure (n = 1079). Also check the miner error while copying the file (Identification heading boxes).

Response: We edited the text. Figure 5 now has the correct number ( n = 1079), and we edited Figure 5 for improved layout and to address style errors. 

  1. In figure- 6, 9, the horizontal or vertical levels are overlapping.

Response: We replaced Figures 6 and 9 and corrected this error.

  1. For all figures, references/permission needs to be presented in figure legends if they are extracted from the other journals. If the figures are created, mention the software from where they are created in the legends. In figure 2, spelling checks of “cell membrane destroied”. Please write full name of MNP in legend. It would be better to demonstrate how the metal nanoparticle are extracted from cells of the body.

Response: No permissions are needed since these figures were all created by us. We edited Figure 2 and its legend.

  1. The mono- and bi-metallic nanoparticle was well discussed. It would be great if they can include about trimetallic nanoparticles too.

Response: Tri-metallic particles are now included, e.g. zinc-copper-iron nanoparticles in the manuscript and in Table 1.

  1. Which software is used for the meta-analysis network? Mention in the figure legend or text.

Response: We now mention in the text and in the relevant figure legends that we used VOS Viewer.

  1. The information in sections 2.6 and 2.6.1 (line 438-469) can be summarized and integrated in the final conclusion.

Response: We revised the text to incorporate that material into the conclusions on page 19.

  1. The authors should make the uniformity in the graphs. Figure. 9 look different as compared to Figure. 6-8

Response: We revised Figure 9 to match the style of the earlier Figure 6.

Reviewer 3 Report

Comments and Suggestions for Authors

Authors collected the information for the said review is complete and with flow as required for the review articles. After careful review, I have noticed that authors should work on following points to improve further.

1.     The bacterial pathogens should be indicated with culture collection details and it is required to mention their ability to produce strong biofilms. For example, P. aeruginosa panel of strains PA01 etc.

2.     It is very difficult to understand that the presented data is all about biofilm formation inhibition or eradication of already formed biofilms. Please provide clarity about this in text.

3.     Table should be revised with above details and time required to inhibit biofilm formation or eradication.

4.     The collected literature showed how surely they confirmed biofilm inhibition and eradication? Is there any data on estimation of EPS/proteins and eDNA?

Other comments:

1.     Title: are authors are sure about the “metal particles size?” the literature compiled here is all about nano? If not, then change to “particles” only.

2.     Line 13: planktonic cells.

3.     Specify the MDR and or strong biofilm producers to the given strains. Give strain identifiers like culture collection number details.

Comments on the Quality of English Language

Minor editing

Author Response

Reviewer 3:

Authors collected the information for the said review is complete and with flow as required for the review articles. After careful review, I have noticed that authors should work on following points to improve further.

  1. The bacterial pathogens should be indicated with culture collection details and it is required to mention their ability to produce strong biofilms. For example, P. aeruginosa panel of strains PA01 etc.

Response: Table 1 now includes ATCC strain information.

  1. It is very difficult to understand that the presented data is all about biofilm formation inhibition or eradication of already formed biofilms. Please provide clarity about this in text.

Response: We reworded the text throughout the paper to emphasise these two different elements. 

  1. Table should be revised with above details and time required to inhibit biofilm formation or eradication.

Response: We reworded the table to clarify the point on biofilm formation versus biofilm eradication. We did not include time because this was not reported uniformly in the studies that were included.

  1. The collected literature showed how surely they confirmed biofilm inhibition and eradication? Is there any data on estimation of EPS/proteins and eDNA?

Response: We checked the included studies listed in the table and in their introduction, these stated the mechanism involved nanoparticles interacting with EPS to cause biofilm inhibition and eradication of existing biofilms. However, these papers did not give data on direct interactions with EPS, protein and eDNA, so we are unable to include those aspects.

Other comments:

  1. Title: are authors are sure about the “metal particles size?” the literature compiled here is all about nano? If not, then change to “particles” only.

Response: All included studies are concerning nano size particles so we clarified that in the text.

  1. Line 13: planktonic cells.

Response: We changed this to read planktonic cells, as suggested.

  1. Specify the MDR and or strong biofilm producers to the given strains. Give strain identifiers like culture collection number details.

Response: We have included ATCC strain identification details in Table 1.

Round 2

Reviewer 1 Report

Comments and Suggestions for Authors

Thank you so much for consideration of all comments. I’m basically pleased with your corrections. I recommended this work can be accepted.

Reviewer 3 Report

Comments and Suggestions for Authors

I congratulate authors for good presentation.